# Enhancement of Electrochemical Detection of Gluten with Surface Modification Based on Molecularly Imprinted Polymers Combined with Superparamagnetic Iron Oxide Nanoparticles

**DOI:** 10.3390/polym14010091

**Published:** 2021-12-27

**Authors:** Dalawan Limthin, Piyawan Leepheng, Annop Klamchuen, Darinee Phromyothin

**Affiliations:** 1College of Materials Innovation and Technology, King Mongkut’s Institute of Technology Ladkrabang, Bangkok 10520, Thailand; 63607009@kmitl.ac.th (D.L.); mildpiyawan55@gmail.com (P.L.); 2National Nanotechnology Center, National Science and Technology Development Agency, Patumthani 12120, Thailand; annop@nanotec.or.th

**Keywords:** gluten, magnetic molecularly imprinted polymers, electrochemical analysis

## Abstract

Novel molecularly imprinted polymers (MIPs) represent a selectively recognized technique for electrochemical detection design. This rapid and simple method prepared via chemical synthesis consists of a monomer crosslinked with an initiator, whereas low sensitivity remains a drawback. Nanomaterials can improve charge transfer for MIP surface modification in order to overcome this problem. SPIONs have semiconductor and superparamagnetic properties that can enhance carrier mobility, causing high sensitivity of electrochemical detection. In this work, surface modification was achieved with a combination of MIP and SPIONs for gluten detection. The SPIONs were synthesized via the chemical co-precipitation method and mixed with MIPs by polymerizing gluten and methyl methacrylate (MMA), presented as a template and a monomer. Magnetic MIP (MMIP) was modified on a carbon-plate electrode. The morphology of modified electrode surfaces was determined by scanning electron microscopy–energy-dispersive X-ray spectrometry. The performance of the MMIP electrode was confirmed by cyclic voltammetry, amperometry, and electrochemical impedance spectroscopy. The MMIP electrode for gluten detection shows a dynamic linear range of 5–50 ppm, with a correlation coefficient of 0.994 and a low detection limit of 1.50 ppm, which is less than the U.S. Food and Drug Administration requirements (20 ppm); moreover, it exhibits excellent selectivity, sensitivity, stability, and reproducibility.

## 1. Introduction

The development of novel sensors to precisely distinguish biochemical molecules or substances has gained significant attention in the food and beverage industries, as well as in clinical diagnostics. Each biochemical molecule distinctly exhibits a different size, shape, functional group, and chemical reactivity [1,2]. In recent years, many specific detection methods have been developed—for example, using biological agents such as enzymes or antibodies, which are known for their high specificity. However, these compounds have certain drawbacks, such as low stability and high cost. On the other hand, molecularly imprinted polymers (MIPs) represent a novel technique for the functional sensing element of electrochemical detection; they recognize the template molecule with specificity based on the cavities in terms of size, shape, and functional group [3,4]. Therefore, this technique has been successfully applied in many fields, such as drug analysis, organic molecule detection, and biomolecule detection, because of its various advantages, including high selectivity, long-term stability, easy preparation, inexpensive synthesis, and facile integration with the carbon-plate electrode [5,6]. Moreover, MIPs have been developed as convenient and portable devices [7]. Molecularly imprinted polymers are synthesized with four basic elements, containing functional monomers, templates, crosslinkers, and initiators. Covalent or non-covalent interaction takes place between a functional monomer and target molecules. The polymer complex forms a three-dimensional network with a crosslinking agent—for example, ethylene glycol dimethacrylate, which can crosslink with the ester functionalities of ethylene glycol dimethacrylate [8,9,10]. The common monomers used for MIP polymerization include methyl methacrylate, 2-vinyl pyridine, styrene, and polyaniline [11,12,13]. Methyl methacrylate (MMA) is a good choice for MIP polymerization, because it consists of small molecules, functional polymers, and is easy to prepare and inexpensive. In contrast, other monomers are large molecules that affect a more non-specific interaction site, causing low selectivity [14]. However, electrode modification with MMA is limited by its insulator properties. The polymer appears to hinder charge transfers between the electrode and the solution, leading to low sensitivity [15]. To solve this problem, molecularly imprinted polymers (MIPs) have been developed with nanomaterial such as gold, silver, multiwalled carbon nanotubes (MWCNTs), and superparamagnetic iron oxide nanoparticles (SPIONs), which have excellent electrical conductivity [16,17,18]. Comparison of SPIONs with gold and silver (noble metal) nanoparticles or MWCNTs shows that the latter are expensive, complicated to prepare, unstable, and toxic, while the SPIONs have many advantages, such as easy preparation, low cost, and biocompatibility [19].

The magnetic field causes the magnetic spin to align when the SPIONs are applied. The SPIONs exhibit giant magnetoresistance (GMR), which leads to a drop in the electrical resistance [20,21,22]. Thus, molecularly imprinted polymers combined with SPIONs show promise in improving charge transferability. This causes high sensitivity of electrochemical detection by MMIPs [23,24]. Many MMIPs have been used to model electrochemical detection [18,19,20,21,25,26,27,28]. MMIP-modified electrodes have been applied to the screening tests of contaminants such as allergens, wherein they can ensure selectivity with the template molecule.

Gluten is a protein found in some rice and flour, and usually used as a component in food and bakery products. Tiny amounts of gluten in the diet may cause enormous gluten allergy symptoms. The U.S. Food and Drug Administration (FDA) requirements state that gluten-free products should contain less than 20 ppm of gluten [28,29,30]. Thus, low-level gluten determination in foods has attracted great attention.

In the present work, MMIP-based electrochemical detection was developed for the sensitive and selective detection of gluten by the co-precipitation processes of synthesized SPIONs and combined MIP–gluten via the polymerization process. Methyl methacrylate (MMA) was used as a monomer, constituting a functional group with gluten as a template. Ethylene glycol dimethacrylate (EGDMA) and 2,2-azobisisobutyonnitrile (AIBN) were used as a crosslinker and an initiator, respectively. Then, MMIP–gluten was modified on a carbon-plate electrode. MMIP–gluten-, MIP-, and NIP-modified electrodes were tested with an electrochemical technique via cyclic voltammetry, amperometry, and impedance spectroscopy. Finally, MMIP–gluten-modified electrodes were analyzed for sensitivity, selectivity, and low detection limit. MMIP–gluten-based electrochemical detection can be developed for the food and beverage industries, among others.

## 2. Materials and Methods

### 2.1. Chemicals and Apparatuses

Gluten, glycine, glutamic acid, phosphate-buffered saline (PBS), methyl methacrylate (MMA), ethylene glycol dimethacrylate (EGDMA), 2,2-azobisisobutyonnitrile (AIBN), iron(III) chloride (FeCl_3_), and polyvinyl alcohol (PVA) were purchased from Sigma-Aldrich, Singapore. Iron(II) chloride (FeCl_2_) and sodium hydroxide (NaOH) were purchased from Carlos (Val de Reuil, France). Six cracker types for real sample analysis were obtained from a local market (Bangkok, Thailand). 

Electrochemical measurements were performed with a µSTAT 400 potentiostat (DropSens, (Llanera, Spain), controlled by DropView 8400 software, connected to a personal computer at room temperature. A triple-electrode system was employed with 3 mm diameter carbon as a working electrode and silver/silver chloride as a reference electrode material. The surface morphology of the MMIP was characterized by field-emission scanning electron microscopy (JEOL-JSM-7600F Schotty FESEM, Tokyo, Japan). 

### 2.2. Preparation of Electrode Modification

Magnetic molecularly imprinted polymer–gluten (MMIP–gluten) preparation was carried out by following procedure in Figure 1. MMIP–gluten was prepared via polymerization, and was composed of gluten, methyl methacrylate (MMA), ethylene glycol dimethacrylate (EGDMA), and 2,2-azobisisobutyonnitrile (AIBN), presented as a template, monomer, crosslinker, and initiator, respectively. 

Firstly, SPIONs were synthesized via a chemical co-precipitation method prepared as described in previous literature [25], where 10 mg of gluten was dissolved in 10 mL of 0.1 M phosphate buffer solution, and 0.53 mL of MMA was dissolved in 2.5 mL of chloroform. Next, 1 mL of 5 mg/mL SPIONs and 1 mL of gluten were added to the mixture and stirred at 60 degrees Celsius. After that, 4.71 mL of EDGMA and 0.47 mL of AIBN were added to the mixture and stirred at 60 degrees Celsius for 30 min. Then, the surface electrode was modified with an MMIP–gluten mixture; 50 μL of the MMIP–gluten mixture was dropped on a working electrode. The surface-modified electrode was controlled via a spin-coating technique at 250 rpm for 30 s. Then, the MMIP–gluten electrode was heated at 60 degrees Celsius for 30 min. Finally, the surface-modified electrode was washed using deionized water and ethanol several times to elute the template. The following surface-modified electrodes were compared: a carbon-plate electrode, a non-imprinted polymer electrode, an imprinted polymer electrode, a magnetic polymer electrode, and a magnetic molecularly imprinted polymer electrode. 

### 2.3. Electrochemical Measurements

Surface-modified electrodes were investigated for gluten detection by using electrochemical measurement. Cyclic voltammetry was performed at a scan range from −1.4 V to −0.2 V and a scan rate of 50 mV/s. Cyclic voltammetry was used to study the potential of the working electrode and to measure the resulting current. A fixed potential was tested via amperometry for 300 s, and presented the measuring current changes. Impedance spectroscopy was performed with a fixed potential and at a measuring frequency range of 1 kHz to 10 MHz. Impedance spectroscopy was also used to study the system and circuit fitting resistance, which were presented via both Bode and Nyquist plots. All sample solutions were dissolved in a 0.1 M phosphate buffer solution for testing. Surface-modified electrodes were tested in six samples for selective analysis. The external magnetic field used was 0.18 tesla for all determined processes.

## 3. Results and Discussion

### 3.1. Surface-Modified Electrode Characterization

Surface-modified electrodes were investigated for gluten detection; NIP, MIP, and MMIP were tested before removing the template, and MMIP was also tested after removing the template. Gluten as a template presented the formation of specific recognition sites after its removal. Furthermore, SPIONs were used as a matrix compound of MIP. The morphology of the surface-modified electrode was characterized by a field-emission scanning electron microscope (FE-SEM). In Figure 2, the FE-SEM image presents the morphology of the surface when the bare electrode was a carbon-plate electrode. Then, the electrode was modified with NIP and MMIP before being removed to present a smooth and flat surface (Appendix A). In contrast, the MIP- and gluten–MMIP-modified electrodes were eluted with gluten templates; the morphology observed in the cavity and the rough MMIP surface are shown in Figure 2a. Moreover, EDS analysis indicated the dispersity of iron and oxygen elements on-MMIP modified electrodes because of the SPIONs, as shown in Figure 2b,c. Thus, FE-SEM and EDS analysis confirmed the morphology and dispersity of the surface-modified electrode of gluten–MMIP with SPIONs.

### 3.2. Electrochemical Behavior of Modified Electrodes

#### 3.2.1. Modeled Electrode Surface Modification

The modified electrode was examined to determine its electrochemical behavior via cyclic voltammetry at a scan rate of 0.05 VS^−1^ and a scan range of −0.2 to −1.4 V. 

The increasing gluten concentration affected the cyclic voltammetry measurements. A changeable redox peak at −0.87 V may be ascribed to the reduction and oxidation of the phosphate buffer and gluten solution (Appendix A). Amperometry presented the correlation between the current and time of the modified electrodes at a constant potential. The current signal was measured for 300 s/sample at an initial potential of −0.87 V. When gluten molecules were detected, the current signal was changed and saturated at 100 s, as shown in Figure 3a. For comparison, the NIP-modified electrode was found to have a low current signal, which did not change significantly as the gluten concentration increased (Appendix A). On the other hand, the MIP-modified electrode showed a slight shift in current signal with increasing gluten concentration. The MMIP-modified electrode presented a higher current signal than the MIP-modified electrode when gluten detection was performed. In addition, the differential current of the MMIP-modified electrode was significantly increased, because SPIONs acted as a pathway of electron flow.

Moreover, EIS is a technique to study the electrode surface’s capacitance and interfacial electron transfer resistance at surface-modified electrodes. In Figure 3b, EIS presents typical Nyquist plots (imaginary impedance vs. real impedance) of the bare carbon-plate, NIP, MIP, MNIP, and MMIP electrodes.

Furthermore, the Nyquist diagram can be used to fit the equivalent circuit in Figure 3c. The equivalent circuit presented charge-transfer resistance of the modified electrode, consisting of solution resistance, surface-modified resistance, and substrate resistance. 

A typical Nyquist plot indicated that the NIP-modified electrode had the largest diameter. This represented the high charge-transfer resistance of NIP, because the polymer is a non-conductive material, leading to hind electron transfer between the electrolyte and the electrode during analysis. 

The MIP-modified electrode showed a decrease in diameter in the Nyquist plot, indicating that its resistance decreases because of easy electron transfer.

Moreover, imprinted polymer combined with SPIONs exhibited the smallest diameter in the Nyquist plot. This represented the lowest charge-transfer resistance, because SPIONs are a semiconductor material that enhances carrier mobility for charge transfer between the electrolyte and the electrode. The resistance analysis of the modified electrode was carried out via amperometry.

#### 3.2.2. Effect of an External Magnetic Field

The SPIONs were attached to the carbon-plate electrode with imprinted polymer. SPIONs are a magnetite that exhibits superparamagnetic properties, showing magnetic behavior with an external magnetic field applied. The magnetic spin is aligned to the direction of the external magnetic field applied [30]. In Figure 4a, amperometric analysis presents the effect of the force of the applied external magnetic field when the MMIP-modified electrode detected gluten molecules. When no magnetic field was applied, the current was low, whereas when the magnetic field was applied—at both high and medium force—the current was higher than with no magnetic field. However, the differential current of the applied magnetic field—whether high or medium force—was insignificant. 

In Figure 4b, EIS analysis indicates the effect of the magnetic field on charge-transfer resistance. Nyquist plots for no magnetic field found a larger diameter than when the magnetic field was applied. SPIONs acted as semiconductors when the MMIP-modified electrode with no magnetic field showed a decrease in resistance. Nevertheless, SPIONs exhibit magnetic behavior with an external magnetic field applied. The giant magnetoresistance (GMR) property, which was enhanced by electron transfer pathways, led to a drop in the electrical resistance [31]. Thus, the MMIP-modified electrode with the applied magnetic field presented the lowest resistance.

### 3.3. Analytical Performance of the MMIP–Gluten-Modified Electrode

The performance of the MMIP–gluten-modified electrode was assessed via amperometry, as shown in Figure 5a, measuring the gluten concentration from 50 to 1000 ppm in PBS (0.1 mol·L^−1^, pH = 6). The results showed that increasing concentrations of gluten caused the current to decrease. The imprinted polymer was stable and recognized gluten molecules at 1000 ppm. Unfortunately, this caused the gluten molecules to hinder electron transfer between the electrode and the solution, decreasing the recorded current.

In Figure 5b, the calibration curve shows a linear response in the range from 5 to 50 ppm, with a sensitivity of 4.192 mA/ppm. The limit of detection (LOD) of gluten was found to be 1.50 ppm, calculated using the equation LOD = 3.3 (S.D.)/M, as shown in Table 1, where S.D. is the standard deviation of the response and M is the slope of the calibration curve [32]. The FDA requires the gluten levels in products with a gluten-free tag to be less than 20 ppm [29,30]. Therefore, MMIP electrodes for gluten detection can detect concentrations below the FDA’s requirements.

### 3.4. Stability and Reproducibility of the MMIP–Gluten-Modified Electrode

MMIP–gluten was prepared via polymerization. The surface-modified electrode was controlled with a spin-coating technique at 250 rpm for 30 s. Six electrodes were compared for detection, displaying similar values of sensitivity and limit of detection. Thus, this method illustrated the excellent stability of gluten–MMIP electrode fabrication.

The stability of MMIP electrode fabrication is presented in Table 2. The MMIP electrode was measured six times at gluten concentrations of 25, 50, and 500 ppm, and displayed good reproducibility five times, as shown in Figure 6b. However, the sixth measurement changed the current value because of defects in the polymer, such as cracks and swelling.

### 3.5. Selectivity of the MMIP-Modified Electrode

Molecularly imprinted polymers represent a specific technique for the recognition of size, shape, and functional molecules. This research used a modified electrode for gluten detection, as gluten consists of amino acids such as glycine and glutamic acid [38]. The specificity of the surface electrode towards gluten was assessed by testing and comparing its responses to gluten and potential interfering substances, such as glycine and glutamic acid. As a result, the MMIP electrode showed a greater current response for gluten than for glycine or glutamic acid, as shown in Figure 7.

Moreover, the EIS analysis of gluten detection presented larger dimensions, indicating high resistance. The imprinted polymer recognizes gluten molecules by their size, shape, and functional molecule, leading to increased resistance and changes in current. Glycine and glutamic acid have a small structure, which cannot match the MMIP electrode’s cavity. Thus, the MMIP electrode presented specific selectivity to the gluten molecules.

### 3.6. Testing in Real Samples 

The MMIP electrode was applied for gluten detection in the real samples. In addition, the MMIP-modified electrodes were tested in six crackers—gluten-free original, gluten-free-salt and vinegar, gluten free seaweed, gluten-free barbeque, original, and wheat—which were assessed by amperometry and EIS analysis. Gluten-free original, gluten-free salt and vinegar, gluten-free seaweed, and gluten-free barbeque are gluten-free products. In contrast, original and wheat crackers are prepared from wheat flour, which contains gluten. 

As shown in Figure 8a, the MMIP electrode exhibited a high-current gluten detection response with original and wheat crackers. In comparison, gluten-free products presented a low-current response. 

Moreover, EIS analysis illustrated high resistivity with the original and wheat crackers compared to the gluten-free products, as shown in Figure 8b. Therefore, the MMIP electrode was sensitive for gluten detection only in original and wheat crackers. Thus, it can be inferred the MMIP electrode has selectivity for gluten detection.

## 4. Conclusions

A magnetic molecularly imprinted polymer was synthesized via combination of an MIP with SPIONs for gluten detection. MMIP–gluten was modified on a carbon-plate electrode and tested via electrochemical techniques, including cyclic voltammetry, amperometry, and impedance spectroscopy. A linear response was found at 5–50 ppm, with an LOD of 1.50 ppm and a sensitivity of 4.192 mA/ppm. The MMIP–gluten model exhibits excellent selectivity, sensitivity, stability, and reproducibility, and can be used to test for gluten detection in real samples. In addition, this model method can be applied for the design of other detection methods.

## Figures and Tables

**Figure 1 polymers-14-00091-f001:**
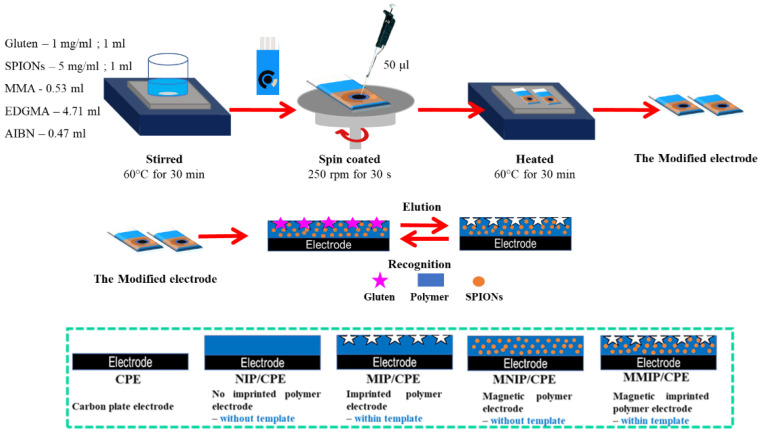
Schematic diagram of the MMIP–gluten electrode preparation and modeled electrode surface modification.

**Figure 2 polymers-14-00091-f002:**
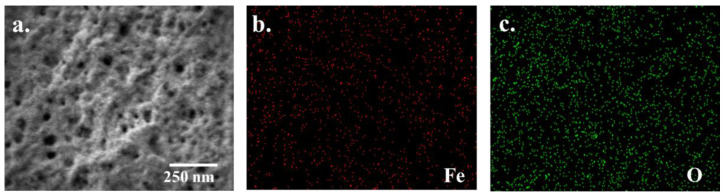
FE-SEM image of the MMIP–gluten electrode (**a**) and EDS images of iron(II) (**b**) and oxygen (**c**).

**Figure 3 polymers-14-00091-f003:**
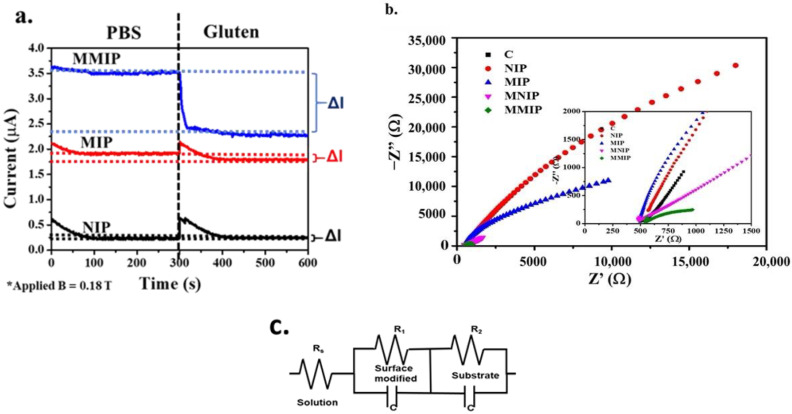
Electrochemical behaviors of MMIP-, MIP-, and NIP-modified electrodes analyzed via amperometry (**a**) and impedance spectroscopy (**b**). EIS circuit of the MMIP electrode (**c**).

**Figure 4 polymers-14-00091-f004:**
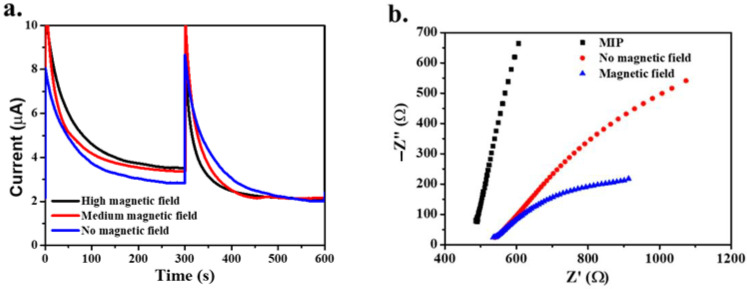
Effect of the applied external magnetic field’s force on current (**a**) and resistance (**b**).

**Figure 5 polymers-14-00091-f005:**
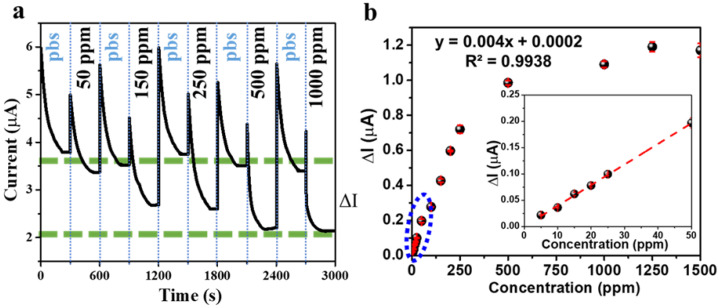
Amperometry responses of the MMIP-modified electrode in the detection of different gluten concentrations (**a**). The calibration curves of gluten, as detected by the MMIP-modified electrode (**b**).

**Figure 6 polymers-14-00091-f006:**
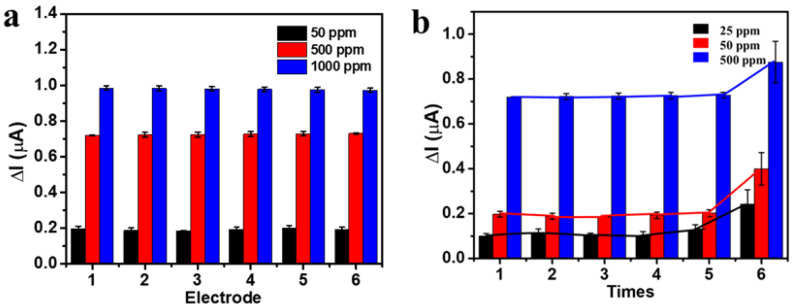
Stability (**a**) and reproducibility (**b**) of the MMIP–gluten-modified electrode.

**Figure 7 polymers-14-00091-f007:**
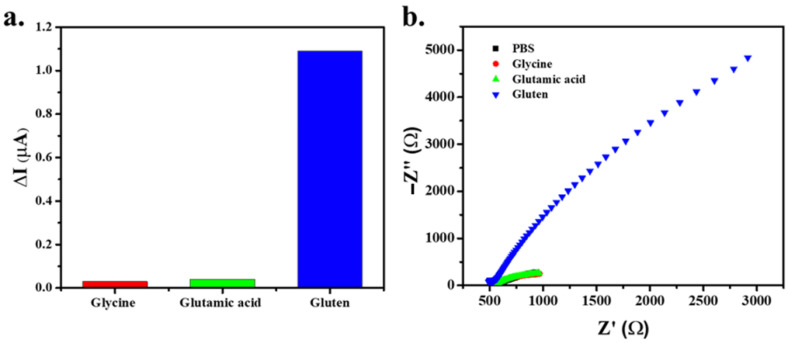
Current (**a**) and resistance response (**b**) of the gluten, as measured by the MMIP-modified electrode.

**Figure 8 polymers-14-00091-f008:**
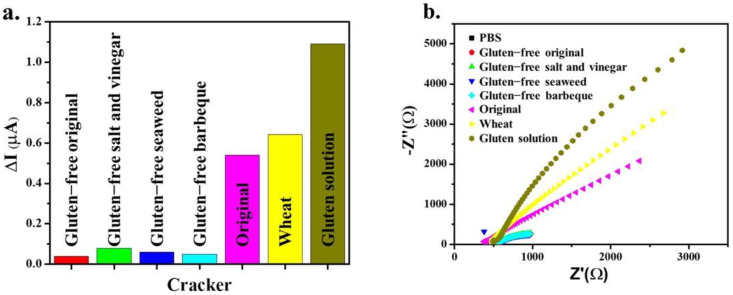
Current (**a**) and resistivity response (**b**) of the gluten in crackers.

**Table 1 polymers-14-00091-t001:** Comparison of the analytical performance of electrochemical assays for gluten detection.

System	Transduction Technique	LOD (ppm)	Dynamic Range (ppm)	Requirement	References
Pencil graphite electrode	Differential pulse voltammetry	7.11	20–100	-	[33]
Aptasensor	Amperometry	0.11	1–100	Low temperature	[34]
Aptasensor	Impedance	5.00	5–50 and 50–1000	Low temperature	[35]
Immunosensor	Amperometry	0.005	0–80	Low temperature	[36]
Aptamer–antibody	Cyclic voltammetry	0.2	0.2–20	Low temperature	[37]
MMIP	Amperometry	1.50	5–1000	-	This work

**Table 2 polymers-14-00091-t002:** Stability of MMIP–gluten electrode fabrication.

Electrode	Linear Equation	R^2^	Sensitivity (mA/ppm)	LOD (ppm)
1	0.004x + 0.0006	0.9925	4.19	1.50
2	0.004x + 0.0001	0.9941	4.21	1.49
3	0.004x + 0.0008	0.9943	4.22	1.53
4	0.004x + 0.0002	0.9932	4.17	1.51
5	0.004x + 0.0005	0.9918	4.19	1.50
6	0.004x + 0.0003	0.9934	4.21	1.52

## Data Availability

The data presented in this study are available on request from the corresponding author.

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
