# Peer review of "Enhancement of Electrochemical Detection of Gluten with Surface Modification Based on Molecularly Imprinted Polymers Combined with Superparamagnetic Iron Oxide Nanoparticles"

_polymers, 2021, doi:10.3390/polym14010091_

Round 1

Reviewer 1 Report

The work presented in this manuscript is novel, interesting, and well-suited for publication in polymers. It reports on the enhanced detection of gluten in foods using molecularly imprinted polymer combined with superparamagnetic iron oxide nanoparticles. There are just a few minor points which need to be addressed before this publication can be accepted.

  1. The introduction is lacking in review of biochemical sensors using different materials. For example, electrochemical detection techniques are commonly combined with conjugated polymers such as polythiophene, polyaniline, or polypyrrole. How does the polymer used in this work compare to such conjugated polymers? Why was PMMA crosslinked with EGDMA used as the polymeric material? Please discuss and expand on the introduction or add the discussion elsewhere in the manuscript, and here is a review you can consider citing (http://dx.doi.org/10.1016/B978-0-12-803581-8.10144-4).
  2. Please combine figures 1 and 2 into a single schematic. In figure 1, the elution step doesn’t seem to fit in the schematic for electrode preparation, it should probably go into figure 2.
  3. In figure 3b, the EDS image for Fe can barely be seen. Please improve on the contrast of the image.
  4. In table 1, how does the sensitivity, LOD and dynamic range of the authors’ gluten sensor compare with other currently existing gluten sensor?

Author Response

Thank you very much for your comment and suggestion. Please see the attachment.

Reviewer 2 Report

This article reports on Enhancement of Gluten-Electrochemical Detection with Surface Modification Based on Molecularly Imprinted Polymers Combined with Superparamagnetic Iron Oxide Nanoparticles. The work is interesting and applied nature and is lying within the scope of polymers. However, the manuscript is very poorly written. Throughout the manuscript the written English is not only grammatically incorrect but also difficult to read and understand what the authors want to communicate. It must be revised extensively and resubmit for consideration in polymers.   

Some examples of poorly written sentences are given below.

Abstract: 

A novel molecularly imprinted polymer (MIP) is a selectively recognized technique for

electrochemical detection design. This rapid and simple method prepared with chemical synthesis

consists of monomer crosslink and initiator, whereas low sensitivity remains a drawback.  

Introduction:

 However, gold, silver, or MWCNT are expensive syntheses, complicated preparation,

instability, and toxicity.

Thus, Low-level detection of gluten is tremendous attention.

Finally, MMIP-gluten was analyzed sensitivity selectivity and low limit of detection.

Materials and Methods

 Magnetic molecularly imprinted polymer-gluten (MMIP-gluten) was the following.

  A fixed potential tested am-perometry for 300 s. which presented the measuring changes in current. 

Cite figure 1 and 2 in the text.

 All determined process was applied 0.18 tesla of the external magnetic field.

Results and Discussion

Very poorly written. Must be revised extensively for readers understanding.

Author Response

Thank you very much for your comment and suggestion. Your opinion is very important to us. Please see the attachment.

Round 2

Reviewer 1 Report

The comments have been comprehensively addressed and the publication is now ready to be accepted in its present form.

Reviewer 2 Report

The quality of the paper has been improved. The revised version is publishable.